# Safety and Efficacy Evaluation of Recombinant Marek’s Disease Virus with REV-LTR

**DOI:** 10.3390/vaccines8030399

**Published:** 2020-07-20

**Authors:** Cuiping Song, Yang Yang, Jing Hu, Shengqing Yu, Yingjie Sun, Xvsheng Qiu, Lei Tan, Chunchun Meng, Ying Liao, Weiwei Liu, Chan Ding

**Affiliations:** Shanghai Veterinary Research Institute, the Chinese Academy of Agricultural Sciences, Ziyue Road 518, Shanghai 200241, China; scp@shvri.ac.cn (C.S.); dcaid@shvri.ac.cn (Y.Y.); hujing0919@shvri.ac.cn (J.H.); yus@shvri.ac.cn (S.Y.); sunyingjie@shvri.ac.cn (Y.S.); xsqiu1981@shvri.ac.cn (X.Q.); tanlei@shvri.ac.cn (L.T.); mengcc@shvri.ac.cn (C.M.); liaoying@shvri.ac.cn (Y.L.); liuweiwei@shvri.ac.cn (W.L.)

**Keywords:** safety, efficacy, recombinant Marek’s disease virus, REV-LTR

## Abstract

Recently, chickens vaccinated with the CVI988/Rispens vaccine showed increased tumor incidence. Moreover, many strains of Marek’s disease virus (MDV) that were naturally integrated with the long terminal repeat (LTR) of the avian reticuloendotheliosis virus (REV) have been isolated, which means it is necessary to develop a new vaccine. In this study, two LTR sequences were inserted into Rispens to construct a recombinant MDV (rMDV). Then, the safety and efficacy of rMDV were evaluated separately in chickens. The growth rate curves showed that the insertion of REV-LTR into MDV enabled a faster replication in vitro than Rispens. Chickens immunized with high or repeated dose rMDV had no MD clinical signs. Further, no tumor, tissue lesions, or evident pathological changes were observed in the chicken organs. Polymerase chain reaction (PCR) and virus isolation revealed that rMDV had the ability to spread horizontally to non-immunized chickens and had no impact on the environment. After five passages in chickens, there were no obvious lesions, and the LTR insertion was stable. There were also no deletions or mutations, which indicates that rMDV is safe in chickens. In addition, rMDV has an advantage over Rispens against vvMDV Md5 at low doses. All results demonstrate that the transgenic strain of rMDV with REV-LTR can be used as a live attenuated vaccine candidate.

## 1. Introduction

Marek’s disease (MD), which is caused by the MD virus (MDV), causes major economic losses to the poultry industry [1]. MDV is divided into three serotypes: type I (MDV-1), type II (MDV-2), and type III (MDV-3) [2], which have a strong ability to spread [3]. The feather follicle epithelium of sick and infected chickens contains large quantities of MDV virions, which could cause environmental pollution [4,5]. Therefore, MDV-infected chickens continuously release infectious virions that, when mixed with dust, spread through the air, increasing the risk of spreading MDV through flocks [5]. 

There are three types of variations in a virus: one caused by mutation of the nucleic acid sequence in the virus replication; one caused by an exchange of the genome segments between different viruses; and one involving gene recombination between viruses in different genera [6]. Recombination can be an essential evolutionary driving force in herpesvirus, and the presence of inverted repeats in the alphaherpesvirus genome allows segment inversion as a consequence of specific recombination between repeated sequences during DNA replication [7]. In the 1990s, Isfort et al. discovered that MDV and reticuloendotheliosis virus (REV) can facilitate natural gene recombination when MDV and REV are serially passaged in chicken embryo fibroblasts (CEFs) [8,9]. Davidson et al. amplified an MDV fragment containing the REV-long terminal repeat (LTR) from tumor samples using the hotspot-combined PCR (HS-cPCR) method, which confirmed that genetic recombination between MDV and REV can also occur in chickens [10]. In China, a large number of epidemiological investigations have found tumors in vaccine-immunized chickens. In 2001, Zhang et al. isolated a natural recombinant MDV wild-type strain, GX0101, integrated with REV-LTR [11]. Because of the low probability of natural recombination between MDV and REV, the recombinant virus can be continuously isolated, indicating that the restructuring process has a competitive advantage [12].

MD can be successfully controlled by vaccination, and CVI988/Rispens is widely used in the intensive production of the poultry industry [13]. With the use of MDV vaccines, the virulence of MDV isolates tends to improve [14,15]. In recent years, tumors have been on the rise even in vaccinated flocks, which means that MD cannot be fully protected by conventional vaccines, especially from supervirulent strains. Thus, a better vaccine is needed. Genetic recombinant vaccines have become a widely researched topic. Using an infectious clone of GX0101, Aijun et al. knocked out REV-LTR through homologous recombination and found that the horizontal transmission ability of GX0101∆LTR was greatly reduced while its pathogenicity was increased, which meant that LTR accelerated the virus’s replication rate and decreased the virulence of the virus [16]. A comparative analysis between a recombinant LTR-inserted MDV and a virulent virus suggested that the horizontal transmission capacity was not necessarily consistent with the pathogenicity of MDV, that is LTR accelerated the virus’s replication but did not change the virulence of the parental virus [17]. Mays et al. found that the insertion of REV-LTR into vvMDV Md5 was fully attenuated in maternal antibody-positive chickens after 40 passages. However, although the vaccination of chicks at hatching with this recombinant virus could protect chickens against MDV-induced bursa and thymic atrophy, it would not provide the same level of protection against MD tumors as Rispens [18]. Previous studies have laid the theoretical foundation to introduce LTR into vaccine strains to construct a new MD vaccine strain. In this study, a recombinant LTR-integrated MDV CVI988/Rispens (rMDV) was constructed, and its safety and efficacy were evaluated to develop a new, effective vaccine against MDV.

## 2. Materials and Methods 

### 2.1. Animals

One-day-old SPF chickens were incubated and kept under controlled temperatures (28 °C–30 °C). The chickens were housed in an isolator with a 12 h light/dark cycle and were given free access to food and water during the study. The care and maintenance of all the animals was done in accordance with the Institutional Animal Care and Use Committee guidelines set by the Shanghai Veterinary Research Institute, Chinese Academy of Agricultural Science (Approval No. SHVRI(SH2015-0118)). 

### 2.2. Viruses

CVI988/Rispens was preserved in a laboratory, and the vvMDV Md5 strain was provided by Professor Cui of Shandong Agriculture University. Viral titers were determined by a viral plaque assay [19]. Viruses were stored in liquid nitrogen until use.

### 2.3. rMDV and Rispens Growth Rate Curves

Chicken embryo fibroblasts (CEF) cells were prepared and cultured for 24 h. Rispens and rMDV were separately diluted at 50–60 PFU/mL and then inoculated onto four plates of CEF cultures. Two plates for each group were harvested at each time point. The virus titer from each culture was determined by a plaque assay. The data used to generate the virus growth rate curves were expressed as the PFU per day/PFU by day 6.

### 2.4. DNA Extraction and PCR

Total DNA was extracted from the tissues of the immunized or control chickens. Briefly, we prepared the pretreatment buffer first, containing 40 μL Tween-20, 40 μL NP40, and 920 μL PBS. Then, we thoroughly mixed 60 μL PBS with 30 μL trypsin K. Next, we took 85 μL tissue grinding fluid in a PCR tube and added 15 μL pretreatment buffer to perform the following procedure: 55 °C for 45 min and 95 °C for 15 min. The DNA extraction was conserved at 4 °C before use.

PCR was carried out in a total volume of 50 μL containing 25 μL of 2× PCR mix (TaKaRa, Japan), 10 μmol/L each of the primers (Table 1), and 2 μL DNA templates. The PCR conditions and cycling parameters were the same for all primer pairs used: denaturation at 94 °C for 2 min, 30 cycles of amplification at 94 °C for 30 s, 55 °C for 30 s, and 68 °C for 3 min, followed by 68 °C for 5 min.

### 2.5. Safety of rMDV

One hundred one-day-old SPF chickens were studied to analyze the safety of rMDV. The chickens were separated into a control group and three virus-inoculated groups. Forty chickens were subcutaneously inoculated with an overdose (10 times the dose) of rMDV containing 30,000 PFU of the virus. The two other groups, with 20 chickens each, were subcutaneously inoculated with one-dose rMDV containing 3000 PFU of the virus and CVI988/Rispens, respectively. These chickens were inoculated 14 days later. The 20 control chickens were subcutaneously injected with phosphate-buffered saline (PBS) (Table 2). All chickens were kept under the same conditions and observed for 60 days.

### 2.6. Horizontal Transmission Capacity of rMDV

Ninety chickens were divided into three large groups (G1, G2, and G3), and each group was divided into two groups (i.e., G1a and G1b; G2a and G2b; and G3a and G3b). Fifteen chickens were then immunized with rMDV, Rispens, or PBS using an overdose (30,000 PFU/dose), and 15 non-immunized chickens cohabited with the immunized chickens (Table 3). On days 7, 14, and 21 after immunization, three chickens were randomly selected from each subgroup. Their blood, feathers, tracheal and rectal swabs, livers, spleens, kidneys, and lungs were collected to detect the specific genes by PCR and to isolate rMDV to investigate the horizontal transmission ability of rMDV. Primers 1 + 4 (Table 1) were used to detect the presence of LTR in IRS. In rMDV, the specific gene was 1730 bp, while that in Rispens was 1088 bp. The CEF cells were inoculated with the treated blood, organs, swabs, feed, water, litter feces, and litter to observe the cytopathic effect (CPE).

### 2.7. Evaluation of the In Vivo Stability of rMDV 

We confirmed the stability of the REV LTR fragment in the recombinant virus during its passage in vivo. Six one-day-old SPF chickens were immunized with rMDV at a 3000 PFU/dose. Seven days later, blood was collected to immunize the six other one-day-old SPF chickens one-by-one for five generations. The last generation was isolated and reared for 60 days. On the 60th day, the feathers were collected, and genomic DNA was extracted. Primers 1 + 2 were used to confirm that the virus was a recombinant virus, and primers 1 + 3 and 1 + 4 were used to identify the presence of LTR in terminal repeat sequence (TRS) and intrinsic repeat sequence (IRS) (Table 1). On the 60th day, visceral tissue, such as that from the liver, spleen, and bursa, were collected for histological examination.

### 2.8. Distribution and Detoxification of rMDV 

A safety experiment was conducted to detect the distribution and detoxification of the recombinant virus in the organs of the chickens. Five chickens were culled on days 3, 7, 21, 42, and 60 after inoculation in the overdose groups. Their blood, feathers, tracheal and rectal swabs, livers, spleens, kidneys, and lungs were collected to detect rMDV by PCR and isolate rMDV. Meanwhile, the air, feed, water, feces, and litter of the breeding environment were also obtained. 

### 2.9. Comparison of the Protective Effects between rMDV and Rispens

SPF chickens that were 61 days old were randomly divided into six experimental groups (G1–G6). Chickens in G1–G3 were immunized with rMDV, and those in G4, G5, and G6 were immunized with Rispens containing 125 PFU/0.2 mL, 250 PFU/0.2 mL, and 500 PFU/0.2 mL, respectively. Twenty chickens were used as controls (control group [GC]) (Table 4). Seven days after vaccination, the chickens were challenged with 500 PFU vvMDV Md5 to observe the protective effects of rMDV and Rispens. All chickens were kept under the same conditions and observed for 60 days.

### 2.10. Pathological Identification and Histological Examination 

In the safety experiment, the clinical signs of each group were recorded after MDV infection. All experimental animals were euthanized and their hearts, livers, spleens, lungs, kidneys, glandular stomach, bursa, and intestines were removed for histological analysis, which was performed as described elsewhere. After 24 h of fixation in 4% formalin, the samples were dehydrated using graded alcohol, embedded in paraffin, and sliced into 4 μm-thick sections. These sections were stained with hematoxylin and eosin and observed under a light microscope (Eclipse TS100; Nikon, Tokyo, Japan).

### 2.11. Evaluation Standard

Chickens with MD clinical symptoms, those with ocular lesions, and those that had no observable lesions but presented pathological changes during the experiment were deemed positive. Chickens that had no clinical observations, no gross lesions, and no pathological changes were considered negative. The suspected samples were subject to confirmation by laboratory tests.

MD incidence criteria was as follows: experimental chicken death (excluding non-specific death), severe weight loss, paralysis syndrome, thymus and bursa Fabricius atrophy, diffuse enlargement of internal organs (including liver, kidneys, heart, spleen, ovaries, etc.) or tumors, peripheral nerve stripes, and swelling signified the onset of MD.

### 2.12. Statistical Analyses

The mean initial weights and mean percentage body weight gains for each group were compared using a one-way ANOVA and Tukey’s multiple comparison test (Prism version 8.0.2). A *p* value ≤ 0.05 was regarded as significant.

## 3. Results 

### 3.1. Construction of rMDV 

Duck embryo fibroblasts (DEFs) were co-infected with MDV (JM/102 W strain) and REV (CSC strain), and the cell suspension was inoculated into the chickens. Finally, a recombinant MDV strain containing LTR was isolated in the chicken feather follicles and named the RM1 virus [20].

RM1, containing the LTR gene fragment, was ligated into the MDV homology arm B40 plasmid to construct a shuttle plasmid containing LTR, namely B40-RM1 Pac. The shuttle plasmid B40-RM1 Pac and the CVI988/Rispens virus genomic DNA was co-transfected into the CEFs, and homologous recombination was performed in the cells to produce rMDV integrated with LTR [21].

### 3.2. In Vitro Replication of the Recombinant Virus 

The growth rate curves of rMDV and the parental virus Rispens infecting the CEFs were established to determine the replication of rMDV. As shown in Figure 1, the Rispens began to form plaques on the second day. It replicated at a uniform rate starting on the third day and peaked on the sixth day. However, compared with the parental strain, during the first three days rMDV had fewer plaques. On the fourth day, it replicated significantly and the replication rate accelerated rapidly on the fifth day. This rapid in vitro replication of the REV-LTR insertion in MDV continued until the seventh day. 

### 3.3. Clinical Signs, Histopathological Analysis, and Weight Gains of Safety Experiment for rMDV 

The clinical signs and histopathological analysis of the immunized chickens were examined in experiment 1 to determine the safety of rMDV in chickens. From the results, we can see that all chickens were well, and no MD clinical symptoms appeared. No tumors or tissue lesions were observed in any of the chicken organs (Figure 2A), and no obvious pathological changes appeared (Figure 2B). In addition, there was no significant difference in the weight gains and organ weights among the four groups after vaccination (Figure 2C). Therefore, regardless of whether overdose or repeated immunization was performed, rMDV was safe for the chickens.

### 3.4. Detection of rMDV in Various Organs of the Immunized Chickens

Experiment 2 was conducted to evaluate the persistence and detoxification of rMDV in the immunized chickens. Five chickens were killed each day on days 3, 7, 21, 42, and 60 after immunization with a high-dose of rMDV. The results showed that the target fragment (1730 bp) could be detected in the organs, blood, and feathers within three days until 60 days after inoculation but could not be detected in the swabs and breeding environment (Figure 3A). The virus isolation results were consistent with the PCR findings. CPE could be observed in the inoculated chicken’s tissues, blood, and feathers, but there was no CPE in the swabs, water, or feed even after one passage (Table 5), which means that rMDV was safe for the environment.

### 3.5. Horizontal Transmission Capacity of rMDV 

In experiment 3, the blood and visceral tissues (namely, swabs, livers, spleens, kidneys, and lungs) were collected from three groups of immunized and cohabiting chickens to detect the horizontal transmission capacity of rMDV. Primers 1 + 4 were specifically used to identify rMDV. The results show that seven days after cohabitation, the target gene (1730 bp) could be amplified in the rMDV-immunized chickens and domestic chickens, while that at 1088 bp could be amplified in the blood and visceral tissues (but not the swabs) of the Rispens-immunized and domestic chickens (Figure 4). Virus isolation results were consistent with the PCR findings (Table 6), which means that rMDV could spread horizontally to non-immunized domestic chickens.

### 3.6. In Vivo Stability of rMDV 

A passage test was performed among the chickens in experiment 4 to estimate the stability of rMDV. In the first four generations, the blood of each chicken was detected by primers 1 + 4 (Figure 5A) to determine the recombinant virus. During the fifth generation of passage, genomic DNA was extracted from the feathers of each chicken and identified with primers 1 + 2, 1 + 3, and 1 + 4 (Figure 5B). The PCR results showed that the target fragments (827 bp, 1448 bp, and 1730 bp) could be detected in each generation of rMDV. The sequencing results of the PCR fragments showed that the target gene (LTR) existed in IRS and TRS without mutation and deletion. There were no obvious lesions or pathological changes in the fifth generation of chickens (Figure 5C). Thus, the LTR insertion was stable in rMDV, which further indicates that rMDV is safe for chickens.

### 3.7. Advantage of rMDV over Rispens against Md5 at Low Doses

A comparative experiment was performed to compare the protective efficacy of rMDV and Rispens. In experiment 5, the two viruses were inoculated subcutaneously into one-day-old SPF chickens at three different doses (125 PFU/0.2 mL, 250 PFU/0.2 mL, and 500 PFU/0.2 mL). On the seventh day after immunization, all chickens were challenged with 500 PFU Md5, and their clinical signs and anatomy were examined. From day 14, MDV-typical symptoms, such as a feathery mess, lassitude, splits, and loss of appetite, began to appear only in the GC group (control) (Figure 6A(a,b)), and only eight birds survived to the end of the experiment (Figure 6B). The autopsy results showed that three of the surviving chickens in the GC group had enlarged kidneys (Figure 6C(a)) and splenomegaly (Figure 6C(c)), and one chicken had testicular swelling (Figure 6C(b)), but no tumors were detected. The chickens in G3 and G6 were in a good mental state and had normal appetite with clean, shiny feathers (Figure 6A(c,d)). There were no suspected tumors in the surviving chickens. One chicken each in G2 and G5 had symptoms, but no death occurred until the end of the test. Two chickens each in G1 and G4 showed symptoms after the challenge. By the end of the experiment, no chicken had died in G1, but two sick chickens in G4 had perished. In the G1 group, ten chickens, including two diseased ones, showed no pathological changes in their organs. In G4, one of the eight surviving chickens was found to have a diffused tumor in the liver (Figure 6C(d)). For 125 PFU immunity, the rMDV and Rispens protection indexes were 77.78% and 66.67%, respectively (Table 7). Thus, at low doses rMDV shows an advantage over Rispens in attacking virulent MDV.

## 4. Discussion

MD is an infectious disease characterized by lymphoid hyperplasia, peripheral nerve paralysis, and tumors in the organs, muscles, and skin of chickens.

Currently, the vaccines for MD are the attenuated serotype I virus vaccine, non-virulent serotype II virus vaccine, and serotype III virus strain HVT. Serotype I MDV vaccines, such as Md11/75 [22] and CVI988/Rispens [23], are mostly produced by virulent strains or medium virulent strains purified by plaque and passaged to reduce their virulence and retain immunogenicity [14]. Serotype II MDV vaccines, such as SB1 [24] and Z4, are isolated from chickens. They are non-pathogenic and tumorigenic but have good immunogenicity [25]. Serotype III MDV vaccines, such as the turkey herpes virus (HVT)-FC126 strain [26,27], were the earliest introduced commercial vaccines for the prevention of MD in China. They are also among the most widely used vaccines for the prevention and control of MD around the world. Multivalent vaccines for MDV currently represent the main promotion and application of bivalent vaccines in the international market, such as the bivalent live vaccine CVI988/Rispens strain + HVT FC126 strain produced by Merria [28]. In addition, trivalent vaccines [29] such as Md11/75 + SB1 + HVT, which is more effective than single-valent vaccines, have been developed internationally. In recent years, conventional vaccines could not produce sufficient protection against MDV because of coinfection with other viruses, including the avian leukemia virus (ALV) [30] and REV [31]. Various methods have been used to construct candidate vaccines for MDV. Genetically engineering vaccines has since become a widely researched topic and breakthrough point. 

REV and MDV are avian diseases that can cause tumors in chickens, and co-infection with MDV and REV is very common in China [32,33]. More than ten years ago, researchers discovered that REV-LTR can be integrated into the genome of MDV when MDV is continuously passaged on cells contaminated with REV [31]. The probability of natural recombination between MDV and REV is very low. Nevertheless, in recent years, several strains of MDVs that have naturally integrated LTR in REV have been isolated [10,11,34], suggesting that a recombinant virus containing this sequence may have a certain survival advantage. This finding provides ideas for the development of a new vaccine for MDV [35]. In this study, two LTR sequences were introduced into the classical MDV vaccine strain Rispens to construct a novel rMDV [20]. The insertion of REV-LTR in MDV offered a faster in vitro replication capacity than Rispens. Although the LTR insertion of rMDV was beneficial for virus replication, there were no clinical symptoms of MD, no difference in body weight, and organ weight, and no obvious MD pathological lesions in the immunized chickens. rMDV could also not be detected in the surrounding environment by PCR and virus isolation. Moreover, in the cohabitation test, the virus could be detected from the feathers and tissues of the non-immunized chickens, indicating that the recombinant virus can spread horizontally. In addition, there was no in vivo deletion or mutation in the insertion gene of the fifth generation, indicating the stability of the recombinant virus. This further illustrates that the novel rMDV was as safe as the parent virus (Rispens) for the target chickens and was harmless to the environment. Together, these results suggest that LTR could enhance virus replication but not virus virulence. This provides a guarantee for the use of LTR to improve the development of MDV. 

In the efficacy comparison experiment, on the seventh day after immunization, all chickens were challenged with Md5. The results showed that, for 125 PFU immunity, the rMDV and Rispens protection indexes against Md5 were 77.78% and 66.67%, respectively. For a single vaccine, CVI988/Rispens is currently the best option and can provide good clinical protection against MDV. MDV is a strictly cell-associated virus which makes it difficult to protect against. At low doses, the insertion of REV-LTR into the Rispens virus has an advantage over using Rispens against Md5, meaning that the insertion of LTR could be a good direction for improvement. In the future, multivalent vaccines or combined vaccines could be developed, which will have a major impact on the development of MDV vaccines. 

Therefore, rMDV with integrated REV-LTR is safe for chickens and could assist against vvMDV, which makes it an ideal candidate vaccine strain. 

## 5. Conclusions

Recombinant Marek’s disease virus with REV-LTR display of high security in target chickens and environment and enhanced efficacy against vvMDV than parental virus. Although more animal experiments will be needed to fully prove the safety and efficacy, our results show that rMDV could be developed as a vaccine candidate against MDV. 

## Figures and Tables

**Figure 1 vaccines-08-00399-f001:**
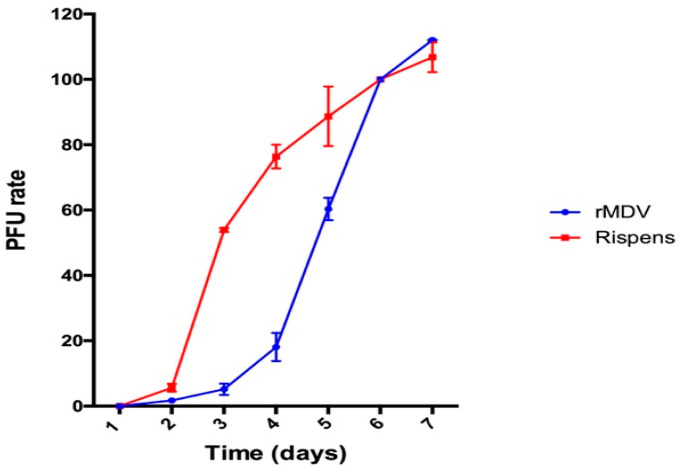
In vitro growth rate of recombinant Marek’s disease virus (rMDV) and Rispens. Rispens and rMDV were separately inoculated onto chicken embryo fibroblast cultures and harvested at each time point. The virus titer from each culture was determined by a plaque assay.

**Figure 2 vaccines-08-00399-f002:**
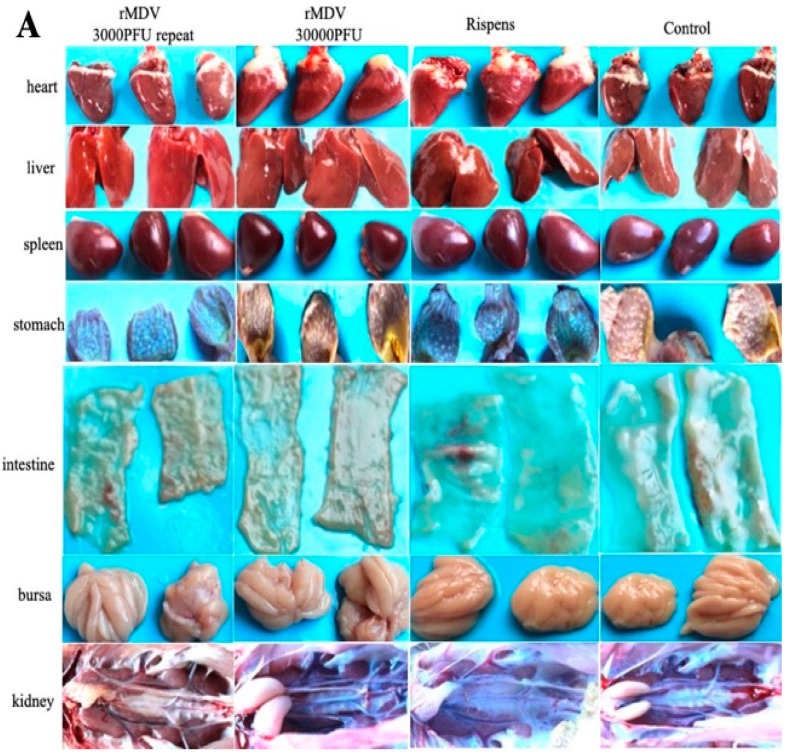
Gross lesions, histopathological analysis and weight gains, and tissue weight in the safety test. (**A**) At day 60 after immunization, the tissues of all chickens were collected to observe their gross lesions. (**B**) At day 60 after immunization, the samples of tissues were fixed in 4% formalin for 24 h and dehydrated, embedded, sliced, and stained with hematoxylin-eosin (HE) for histopathological analysis. (**C**) At each time point, all chickens were weighed to analyze their weight gains, and at day 60 the tissues of each group were collected to analyze the organ weights (a: body weight; b: tissue weight; NS: no significant difference).

**Figure 3 vaccines-08-00399-f003:**
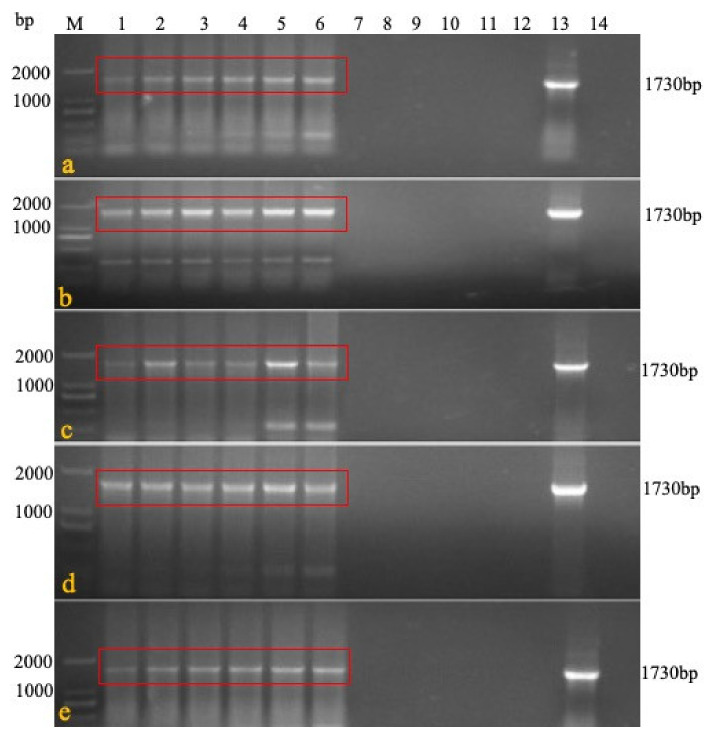
rMDV detection by PCR in the persistence and detoxification test. At each time point, DNA was extracted from the blood, feathers, swabs, feed, feces, litter, water, and tissues of the immunized chickens, and PCR was carried out to detect rMDV (**a**): target sequence 1730bp could be detected in blood and tissues at 3 days after immunization; (**b**): target sequence 1730bp could be detected in blood and tissues at 7 days after immunization; (**c**): 21 days; (**d**): 42 days; (**e**): 60 days; M: Marker 1: blood; 2: feather; 3: liver; 4: spleen; 5: kidney; 6: lung; 7: tracheal swab; 8: rectal swab; 9: feed; 10: feces; 11: litter feces; 12: water; 13: positive; 14: negative).

**Figure 4 vaccines-08-00399-f004:**
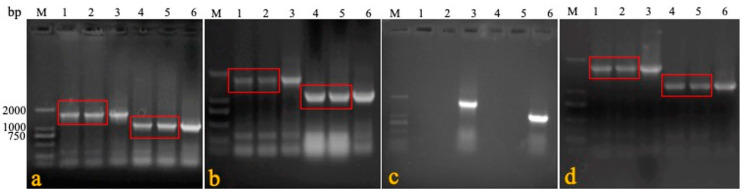
PCR detection in horizontal test. DNA was extracted from the blood, feathers, swabs, and tissues of immunized and cohabiting chickens, and PCR was carried out to determine the horizontal transmission capacity of rMDV (**a**: blood; **b**: feather; **c**: tracheal swab; **d**: liver; **e**: spleen; **f**: kidney; **g**: lung; **h**: rectal swab; M: marker; 1: rMDV-immunized chicken; 2: rMDV cohabiting chicken; 3: rMDV-positive; 4: Rispens-immunized chicken; 5: Rispens cohabiting chicken; 6: Rispens-positive).

**Figure 5 vaccines-08-00399-f005:**
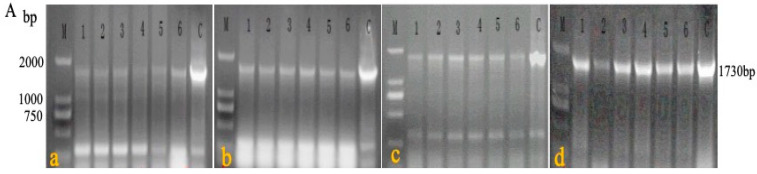
PCR detection and histopathological analysis in the passage test. (**A**) DNA was extracted from the blood of the first four generations, and PCR was carried out using primers 1 + 4 to detect rMDV; the target gene was 1730 bp (**a**: first generation; **b**: second generation; **c**: third generation; **d**: fourth generation; 1, 2, 3, 4, 5, 6: chickens of each generation). (**B**) DNA was extracted from the feathers of the fifth generation, and PCR was carried out using primers 1 + 2, 1 + 3, 1 + 4 to detect rMDV (**a**: rMDV detection by primers 1 + 2, and the target gene was 827 bp; **b**: rMDV detection by primers 1 + 3, and the target gene was 1448 bp; **c**: rMDV detection by primers 1 + 4, and the target gene was 1730 bp). (**C**) On the 60th day, visceral tissues of the fifth generation were collected for histological examination (**a**: heart; **b**: liver; **c**: spleen; **d**: lung; **e**: kidney; **f**: bursa).

**Figure 6 vaccines-08-00399-f006:**
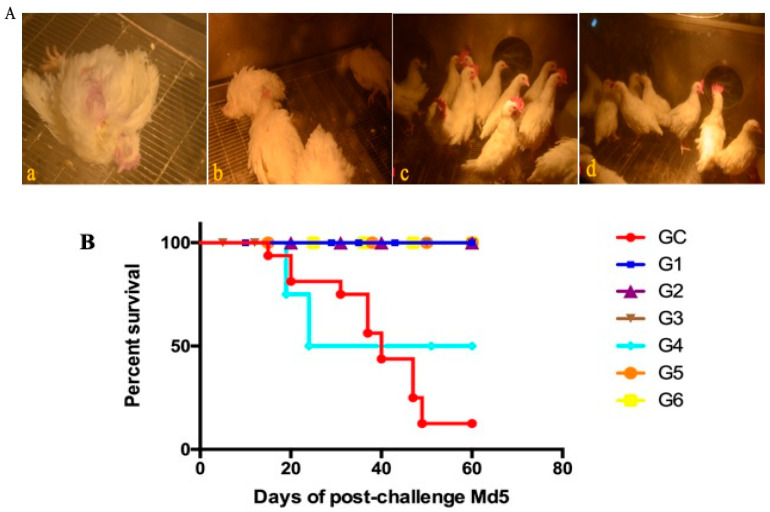
Clinical signs, survival curves, and gross lesions in the comparative experiment. (**A**) Clinical signs were observed after the Md5 challenge in each group ((**a**), (**b**): GC group; (**c**): G3 group; (**d**): G6 group). (**B**) The survival curves of each group were generated after the Md5 challenge. (**C**) Gross lesions of the survived chickens of the control ((**a**): enlarged kidneys; (**b**): testicular swelling; (**c**): splenomegaly) and G4 groups ((**d**): a diffused tumor in the liver).

**Table 1 vaccines-08-00399-t001:** Primers for PCR.

Primer	Sequences (5′–3′)
1	gccctgtcgaagaggaaata
2	ctatttgcgcggaggaag
3	cagccttcgaaatatatctca
4	ccctttatgaaagctggcctc

**Table 2 vaccines-08-00399-t002:** The safety of recombinant MDV (rMDV).

Group		Virus	Inoculation Age (Days)	Chicken Number	Inoculation Dose (PFU)	Inoculation Route	Repeat Inoculation Age
Experiment group	1	rMDV	1	20	3000	Subcutaneous	15
	2	rMDV	1	40	30,000	Subcutaneous	/
	3	Rispens	1	20	3000	Subcutaneous	15
Control	4	Phosphate-buffered saline (PBS)	1	20	/	/	/

**Table 3 vaccines-08-00399-t003:** Horizontal transmission capacity of rMDV.

Group		Virus	Inoculation Age (Days)	Chicken Number	Inoculation Dose (PFU)	Inoculation Route
G1	G1a	rMDV	1	15	30,000	Subcutaneous
	G1b			15		
G2	G2a	Rispens	1	15	30,000	Subcutaneous
	G2b			15		
G3	G3a	PBS	1	15	/	Subcutaneous
	G3b			15		

**Table 4 vaccines-08-00399-t004:** The protective effects of rMDV and Rispens.

Group	Virus	Inoculation Age (Days)	Chicken Number	Inoculation Dose (PFU/0.2 mL)	Challenge Virus	Challenge Age (Days)	Challenge Dose (PFU)
G1	rMDV	1	10	125	Md5	7	500
G2	rMDV	1	10	250	Md5	7	500
G3	rMDV	1	10	500	Md5	7	500
G4	Rispens	1	10	125	Md5	7	500
G5	Rispens	1	10	250	Md5	7	500
G6	Rispens	1	10	500	Md5	7	500
GC	/	/	20	/	Md5	7	500

**Table 5 vaccines-08-00399-t005:** Virus isolation in the persistence and detoxification test.

	3 Days	7 Days	21 Days	42 Days	60 Days
Blood	+	+	+	+	+
Feather	+	+	+	+	+
Liver	+	+	+	+	+
Spleen	+	+	+	+	+
Kidney	+	+	+	+	+
Lung	+	+	+	+	+
Tracheal swabs	-	-	-	-	-
Rectal swabs	-	-	-	-	-
Feed	-	-	-	-	-
Water	-	-	-	-	-
Litter feces	-	-	-	-	-

“+” means cytopathic effect (CPE); “-” means no CPE.

**Table 6 vaccines-08-00399-t006:** Virus isolation in horizontal test.

	rMDV	rMDV Cohabitation	Rispens	Rispens Cohabitation	Control
Blood	+	+	+	+	-
Feather	+	+	+	+	-
Liver	+	+	+	+	-
Spleen	+	+	+	+	-
Kidney	+	+	+	+	-
Lung	+	+	+	+	-
Tracheal swabs	-	-	-	-	-
Rectal swabs	-	-	-	-	-
Feed	-	-	-	-	-
Water	-	-	-	-	-

“+” means CPE; “-” means no CPE.

**Table 7 vaccines-08-00399-t007:** Table of the immune protection index.

Group	Virus	Inoculation Dose	MD Number		MD Positive	MD Incidence (%)	Mortality Rate (%)	PI (%)
			Lesion	Died				
G1	rMDV	125	2/10	0	2/10	20	0	77.78
G2	rMDV	250	1/10	0	1/10	10	0	88.89
G3	rMDV	500	0	0	0	0	0	100
G4	Rispens	125	3/10 *	2/10	3/10	30	20	66.67
G5	Rispens	250	1/10	0	1/10	10	0	88.89
G6	Rispens	500	0	0	0	0	0	100
GC	/	/	18/20 ^#^	14/20	18/20	90	70	/

* Anatomy of a chicken with a liver tumor. ^#^ Anatomy of three chickens with kidney enlargement and one chicken with testicular swelling.

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
