# Peer review of "Safety and Efficacy Evaluation of Recombinant Marek’s Disease Virus with REV-LTR"

_vaccines, 2020, doi:10.3390/vaccines8030399_

Round 1
Reviewer 1 Report
I have attached the author's pdf with a substantial number of textual edits. The use of the English language is poor and I thought the authors could use some additional help. I spent many hours working on editing this manuscript and I believe that the text edits will help to clarify what the authors are trying to express.
Ultimately, the issue of whether your rMDV is better than the Rispens vaccine hinges on the slightly higher protection indices (77.8% versus 66.7%) and the ability to use lower doses of rMDV than Rispens virus. Is this enough to move the industry to switch? Please comment on this.
I made comments on the figures and legends where I thought they could be improved. The multi-panel gross and histology photos are too small to assess. The same is true of the virus isolation panels showing cytopathology on CEF's. Could the virus isolation cytopathology panels be placed in a table that simply scores + or - indicating virus isolation or not? The panels are too small for the reader to evaluate.

Author Response
Dear reviewer:
Thanks for all the valuable comments. These comments help us to improve the quality of our manuscript. We have carefully considered the comments and corresponding corrections have been made. The detailed replies are as follows:
Ponit 1: I have attached the author's pdf with a substantial number of textual edits. The use of the English language is poor and I thought the authors could use some additional help. I spent many hours working on editing this manuscript and I believe that the text edits will help to clarify what the authors are trying to express.
Response 1: We have edited the manuscript use a professional English editing service.
Ponit 2: Ultimately, the issue of whether your rMDV is better than the Rispens vaccine hinges on the slightly higher protection indices (77.8% versus 66.7%) and the ability to use lower doses of rMDV than Rispens virus. Is this enough to move the industry to switch? Please comment on this.
Response 2: Yes. In terms of a single vaccine, CVI988/Rispens is currently the best vaccine, and it can provide good protection for MDV clinically. MDV is a strict cell-associated virus, and it is difficult to improve a little protection effect. At low doses, rMDV shows an advantage over Rispens in attacking virulent MDV, which means that the insertion of LTR could be a good direction for improvement. In the future, multicalent vaccines or combined vaccines can be tried to make which will have a major impact on the development of MDV vaccine.
Ponit 3: I made comments on the figures and legends where I thought they could be improved. The multi-panel gross and histology photos are too small to assess. The same is true of the virus isolation panels showing cytopathology on CEF's. Could the virus isolation cytopathology panels be placed in a table that simply scores + or - indicating virus isolation or not? The panels are too small for the reader to evaluate.
Response 3: All pictures have been re-edited, and a table scores + or – indicating virus isolation or not has been made. Thank you for this suggestion.
Reviewer 2 Report
Ths article by Cuiping Song et al. described very interesting results on the safety and efficacy of a recombinant virus against Marek's Disease, an affliction in poultry industry.
Some modifications have to be done before possible publication in Vaccines.
General comments :
Latin location have to be written in italic
Please verify grammar structure, police choice and space that are missing.
Please introduce all abbreviation in the main text (and not abstract) before their first use.
Number until elven have to be written in full letter, in absence of units.
Major :
Virus plaque assays have to be detailed or a reference has to be added.
Why have the authors not repeated the inoculation of a ten-fold dose? This could optimize the results.
The molecular amplification method have to be completed : -thermocycler program; - reference or in-house primers; - use of negative and positive control; method of extraction for nucleic acids.
For pathological examination, how many operators analysed the biological samples ? why have the authors not used immunostaining?
Minor:
Line 65 and 73 add et al. when referring to an article.
How many thawing before their use (line 95-96)?
Give version detail about the Prism software.
I don't think that poultry picture is useful in the present presentation. Please consider add them in supplementary data.
Figure legend have to be completed adding a brief summary of the pertinent information they provided (especially for non-pathologists readers).
Author Response
Dear reviewer:
Thanks for all the valuable comments. These comments help us to improve the quality of our manuscript. We have carefully considered the comments and corresponding corrections have been made. The detailed replies are as follows:
Ponit 1: Virus plaque assays have to be detailed or a reference has to be added.
Response 1: Yes, we have added the reference.
Ponit 2: Why have the authors not repeated the inoculation of a ten-fold dose? This could optimize the results.
Response 2: Because CVI988/Rispens is only immunized one time clinically, there is no possibility of 10-fold dose inoculation. The Animal Welfare Committee of Shanghai Veterinary Research Institute, Chinese Academy of Agricultural Science believes that from a humanitarian perspective, there is no necessary to repeat the inoculation of a 10-fold dose, so in safety test, a 10-fold dose and a repeated one-dose inoculation were made to determine the safety of rMDV.
Ponit 3: The molecular amplification method have to be completed : -thermocycler program; - reference or in-house primers; - use of negative and positive control; method of extraction for nucleic acids.
Response 3: DNA extraction and PCR have been completed.
Ponit 4: For pathological examination, how many operators analysed the biological samples ? why have the authors not used immunostaining?
Response 4: For pathological examination, two operators have analyzed the biological samples. In safety test and passage test, there were no obvious lesions nor pathological changes in tissues of immunized chickens, then immunostaining could not be detected.
Minor:
Ponit 5: Line 65 and 73 add et al. when referring to an article.
Response 5: Yes, we have added the et al.
Ponit 6: How many thawing before their use (line 95-96)?
Response 6: The virus was thawing in 37°C quickly before use.
Ponit 7: Give version detail about the Prism software.
Response 7: We use Prism version 8.0.2.
Ponit 8: I don't think that poultry picture is useful in the present presentation. Please consider add them in supplementary data.
Response 8: Thanks for the suggestion. We have added the clinical pictures in supplementary data.
Ponit 9: Figure legend have to be completed adding a brief summary of the pertinent information they provided (especially for non-pathologists readers).
Response 9: All pictures have been re-edited, and brief summary have been added in the figure legend.
Reviewer 3 Report
The manuscript by Song et al. seeks to investigate novel vaccines for Mareks disease by comparing the Rispen vaccine to a recombinant one with an insertion of a REV-LTR.
The language used throughout the paper, starting even with the Abstract, made reading this manuscript difficult and there are way too many examples of bad phrasing or unreadable sentences throughout to cite each one.
The figures are also exceptionally bad. There is no stats in figure 1. Figure 2 and 3 and 4 can barely be seen....they are either too small or too dark to make sense of them. They are also missing stats throughout. The figure legends are too short to be useful and thus what the figures are showing are difficult to discern. Figures like the histology also appear somewhat distorted and blurry. I cannot tell if the figures support the results discussed since I cannot seem them. Clinical signs in figure 2 do not seem to support that this vaccine is any better than the Rispen one.
Author Response
Dear reviewer:
Thanks for all the valuable comments. These comments help us to improve the quality of our manuscript. We have carefully considered the comments and corresponding corrections have been made. The detailed replies are as follows:
Ponit 1: The language used throughout the paper, starting even with the Abstract, made reading this manuscript difficult and there are way too many examples of bad phrasing or unreadable sentences throughout to cite each one.
Response 1: We have edited the manuscript use a professional English editing service.
Ponit 2: The figures are also exceptionally bad. There is no stats in figure 1. Figure 2 and 3 and 4 can barely be seen....they are either too small or too dark to make sense of them. They are also missing stats throughout. The figure legends are too short to be useful and thus what the figures are showing are difficult to discern. Figures like the histology also appear somewhat distorted and blurry. I cannot tell if the figures support the results discussed since I cannot seem them. Clinical signs in figure 2 do not seem to support that this vaccine is any better than the Rispen one.
Response 2: All pictures have been re-edited, and brief summary have been added in the figure legend.
Round 2
Reviewer 3 Report
There are still some major English language flaws, especially with tenses or repeating of words within the same sentence throughout the manuscript.
Results do not belong in the figure legends. Simply describe what was done and save the results for the result sections.
I still do not see mentions of statistical analysis in the figures. What was significant and what was not?
Figure quality remains poor. Many figures are out of focus or dark still.
Author Response
Dear reviewer,
Thank you very much for your critical review of our manuscript. We have carefully considered the comments and corresponding corrections have been made. We very much appreciate these comments and believe that the revision has greatly improved the quality of our manuscript that can be accepted for publication in “Vaccines”. The detailed replies are as follows:
Ponit 1: There are still some major English language flaws, especially with tenses or repeating of words within the same sentence throughout the manuscript.
Response 1: The manuscript had been re-edited by MDPI.
Ponit 2: Results do not belong in the figure legends. Simply describe what was done and save the results for the result sections.
Response 2: Brief summary had been added in the figure legends, and the results had been re-described simply.
Ponit 3: I still do not see mentions of statistical analysis in the figures. What was significant and what was not?
Response 3: The statistical analysis had been made in Fig2C, and there was no significant difference in body weight and organ weights in safety experiment.
Ponit 4: Figure quality remains poor. Many figures are out of focus or dark still.
Response 4: The figure 2A, 2B had been re-edited.
Thank you and best regards.
Yours sincerely,
Cuiping Song